# Equivariant Graph Learning for High-density Crowd Trajectories Modeling

**Yang Liu** *yliukj@connect.ust.hk*
*Hong Kong University of Science and Technology*
*Hong Kong University of Science and Technology (Guangzhou)*

**Zinan Zheng** *zzheng078@connect.hkust-gz.edu.cn*
*Hong Kong University of Science and Technology (Guangzhou)*

**Yu Rong** *yu.rong@hotmail.com*
*Alibaba DAMO Academy*

**Jia Li**[*] *jialee@ust.hk*
*Hong Kong University of Science and Technology (Guangzhou)*

**Reviewed on OpenReview:** *https://openreview.net/forum?id=TeQRze2ZjO*

## Abstract

Understanding the high-density crowd dynamics of urbanization plays an important role in architectural design and urban planning, preventing the occurrence of crowd crush. Most traditional methods rely on formulas designed based on expert knowledge, which are inflexible and incomplete to model complex real-world crowd trajectories. To address the issue, recent studies propose to simulate crowds via data-driven models. However, these models fail to learn the inherent symmetry of high-density crowd trajectories, leading to insufficient generalization ability. For example, existing models can not predict left-to-right trajectories by learning right-to-left trajectories, even though they share similar patterns. In this work, we propose a novel Equivariant Graph Learning framework for high-density crowd dynamic modeling, called CrowdEGL. It utilizes an additional objective to encourage models to predict the transformed output given the input under the same transformation. We summarize three types of transformation groups, which are determined by the symmetry of environments. To explicitly incorporate these augmented data, a multi-channel GNN is employed to learn the latent graph embedding of pedestrian patterns. Finally, to model dense crowd interactions, future positions of original and transformed inputs are obtained by multiple independent graph decoders. Extensive experiments on 8 datasets from 5 different environments show that CrowdEGL outperforms existing models by a large margin.

## 1 Introduction

The increasing population and resulting urbanization have led to a significant rise in crowd density, particularly in urban areas such as metro stations, stadiums, and exhibition centers (Sindagi & Patel, 2018). Therefore, understanding high-density crowd dynamics is essential for applications in public safety (Johansson et al., 2008). Specifically, given the initial states (locations, velocities, etc.) of crowds, our objective is to forecast their future state by modeling interactions among pedestrians and the impact of environments. Such forecasting is crucial for crowd management (Xu et al., 2014), architectural design, and urban planning. Otherwise, without proper and sufficient safety measures, the gathering of large crowds in confined spaces can lead to stampede incidents, causing injuries and fatalities. For example, Seoul Halloween crowd crush in 2022 killed 159 people and 196 others were injured (contributors, 2023); In 2023 Freedom City Mall crush, crowds rushing to see a firework display got stuck in a narrow corridor and killed 10 people (Atuhaire, 2023).

Figure 1: (a) Illustration of high-density crowds at a 90°crossing, which contains dense interactions; (b) Rotation equivariance of pedestrian trajectory. They enter through the four entrances to reach the opposite exits. Trajectories starting from different entrances are indicated by different colors.

Early crowd modeling approaches (Yang et al., 2020) rely on formulas designed based on *human prior knowledge*. Among them, the Boids algorithm (Reynolds, 1987) is a representative model, which simulates crowd dynamics through three fundamental rules: separation, alignment, and cohesion. Another widely used method is the social force model (Helbing & Molnar, 1995) which characterizes interactions among individuals, obstacles, and destinations as different types of forces (repulsive or attractive) and utilizes Newtonian mechanics to update their behaviors. However, these assumptions are incomplete and overly restrictive to model real-world crowd dynamics, which are inherently complex and heterogeneous (Moussaïd et al., 2011). Therefore, recent works (Shi et al., 2023; Zhang et al., 2022; Yu et al., 2023) have employed *data-driven* models to obtain more realistic crowd trajectories and demonstrate promising performance. In particular, they treat individuals, obstacles, and destinations as nodes and construct graphs based on their distance. Graph Neural Networks (GNNs) (Gilmer et al., 2017) are then employed to encode their interactions, and a Multilayer Perceptron (MLP) decoder is commonly used to predict the future states of the crowd.

Nevertheless, current GNNs (Shi et al., 2023; Sanchez-Gonzalez et al., 2020; Gilmer et al., 2017) are insufficient to capture internal symmetries of the physical world (Cohen & Welling, 2016; Walters et al., 2021; Xu et al., 2023), thus they fail to generalize in different directions. Figure 1 displays high-density crowd trajectories in a 90°crossing. Pedestrians enter the crossing from four entrances and aim to exit from the opposite exits. In such highly crowded situations, we can observe that pedestrian trajectories demonstrate specific rotational symmetries. For instance, if the initial states of pedestrians are rotated by 180°, the crowd trajectories would rotate in an analogous way, known as rotation equivariance. With reduced individual freedom of movement, the interactions among pedestrians, as well as the interactions between pedestrians and environments, are nearly invariant to such rotations. Regardless of this inductive bias, GNNs are inadequate to learn the real crowd dynamics and tend to overfit the observed trajectories (See Figure 5).

Although multiple GNNs (Satorras et al., 2021; Han et al., 2022a; Huang et al., 2022) have been proposed to predict physical dynamics with strictly equivariant under a given group action, it remains non-trivial to design symmetric crowd dynamic models because of the following challenges: (1) **Dense interactions**. In high-density circumstances, crowd interactions occur with great frequency, so that pedestrians are often forced to take detours or even change their intended destinations (Cao et al., 2017); (2) **Imperfectly symmetric**. Due to the effects of varying human psychology and physiology, they might not react completely the same to nearby pedestrians and the environment, breaking the symmetry of crowd trajectories to a certain degree; (3) **Finite group actions**. Existing equivariant GNNs (Han et al., 2022b) leverage the complete symmetry of the Euclidean space, ensuring their outputs will rotate/translate/reflect in the same way as the inputs. Such constraint is too strong for crowd dynamic modeling (Dangovski et al., 2022; Wang et al., 2022). For instance, in Figure 1, a 100°rotation of crowds would violate the boundary condition and thus be invalid.

In this work, we propose a novel Equivariant Graph Learning framework called CrowdEGL, encouraging GNNs to learn a more practical imperfect equivariance property. Specifically, it achieves this objective via an equivariance loss that is constructed based on data augmentation. It works as a regularization term so that we can control the degree of equivariance. The augmented strategies are defined by cyclic groups (Suzuki,

1986). We summarize them for various environments such as corridors and crossings. For example, for the crowd at the crossing, we use the cyclic group of 90°rotations to describe its rotation equivariance. Furthermore, to incorporate crowd trajectories that exhibit a high degree of similarity in behavior patterns, we explicitly treat augmented data as input and propose a multi-channel message-passing GNN to learn latent graph representations. Finally, to capture dense crowd interactions and allow the model to behave distinctly in different directions, multiple independent GNNs are utilized as decoders. Extensive experiments on five different crowded environments demonstrate that CrowdEGL has a better generalization ability over state-of-the-art models and is beneficial for learning the symmetry of crowd dynamics. Our main contributions are summarized as follows:

- We emphasize the significance of learning high-density crowd dynamics, which is challenging and crucial for applications in human activity and society. We highlight the importance of symmetry for high-density crowd trajectory prediction. To our knowledge, it is the first time equivariance is considered in data-driven crowd simulation.

- We propose a novel and flexible equivariant learning framework to incorporate imperfect equivariance in practical crowd trajectories to backbone GNNs, and design a multi-channel GNN to model common pedestrian behavior patterns in different directions.

- We conduct extensive experiments to evaluate the performance of CrowdEGL on eight datasets of five crowded environments. Experimental results show that it achieves significantly better generalization ability over the state-of-the-art models. Ablation studies demonstrate the effectiveness of our model designs. To facilitate the research on high-density crowd modeling, we make our datasets and implementation available at Supplementary Material.

## 2 Preliminary

### 2.1 Problem Definition

Different from existing studies (Alahi et al., 2016; Gupta et al., 2018; Kothari et al., 2022; Shi et al., 2021; Yu et al., 2020) on urban open space (e.g., parks), in this work, we study trajectories in various bounded and narrow environments that have extremely dense interactions. Therefore, we consider a challenging problem setting that focuses on learning interactions of initial crowd states and forecasting their positions after a fixed time interval. Compared to existing works (Shi et al., 2023; Zhang et al., 2022), we omit historical states and destinations, which are employed to reflect pedestrian intentions but are not always available in practice.

Specifically, as shown in Figure 1, we employ discretized obstacles to represent the environments (e.g., walls). Then pedestrians and obstacles can be treated as nodes in a graph. At time $t$, each node $i$ is represented by: (1) geometric features including the position vector $\boldsymbol{x}_i^{(t)} \in \mathbb{R}^2$ and the velocity vector $\boldsymbol{v}_i^{(t)} \in \mathbb{R}^2$; (2) non-geometric features such as the node type (pedestrian or obstacle), denoted by $\boldsymbol{u}_i$; (3) spatial connection with others, where an edge $e_{ij}$ is constructed via distance cutoff and the edge attributes (e.g., node distances) are denoted by $a_{ij}$. For simplicity, we denote $(\boldsymbol{x}^{(t)}, \boldsymbol{v}^{(t)})$ and $(\boldsymbol{u}, \boldsymbol{e} = \{e_{ij}\}, \boldsymbol{a} = \{a_{ij}\})$ as dynamic crowd states and graph features correspondingly. Formally, the crowd trajectory forecasting problem is defined as follows:

**Definition 2.1** *(Crowd Trajectory Forecasting) Given the initial crowd states $(\boldsymbol{x}^{(t)}, \boldsymbol{v}^{(t)})$ at time $t$ and graph $(\boldsymbol{u}, \boldsymbol{e}, \boldsymbol{a})$, the objective is to predict the subsequent position $\boldsymbol{x}^{(t+\Delta t)}$, where $\Delta t$ is the target time interval.*

### 2.2 Graph Neural Networks

GNNs (Gilmer et al., 2017; Kipf & Welling, 2017; Xu et al., 2019; Zhao et al., 2024; Sun et al., 2023; Tang et al., 2022) are neural models specifically designed to learn graph-structured data with complex relationships and dependencies. Recently, they have shown great potential in simulating physical systems such as fluid (Sanchez-Gonzalez et al., 2020; Li & Farimani, 2022) and molecular dynamics (Huang et al.,

2022). They achieve this through iteratively propagating and updating information through the nodes and edges. Let $\boldsymbol{h}_i^{(l)}$ be the $l$-th layer embedding of node $i$ and $\boldsymbol{m}_{ij}^{(l)}$ denotes the $l$-th layer message embedding between node $i$ and $j$. The $l$-th GNN layer computes:

$$\boldsymbol{m}_i^{(l)} = \mathrm{Agg}(\{\boldsymbol{m}_{ij}^{(l)}\}_{j \in \mathcal{N}_i}), \quad \boldsymbol{h}_i^{(l)} = \mathrm{Combine}(\boldsymbol{h}_i^{(l-1)}, \boldsymbol{m}_i^{(l)}), \tag{1}$$

where $\mathcal{N}_i$ collects the neighbors of node $i$. The common choices of Agg function are Sum or Mean, while that of Combine function is Multi-layer Perceptrons (MLPs). The final prediction is obtained by applying several iterations of message passing. Most existing Equivariant GNNs (e.g., EGNN and GMN) and Graph Neural Simulators such as GNS and CrowdSim are based on such a framework.

### 2.3 Equivariance

To improve model generalization, we force them to be equivariant, which refers to the property where the output of a system changes in the same way as the input changes. Formally, equivariance is defined on a specific group (Suzuki, 1986):

**Definition 2.2** *(Group) A group is a set of operations that satisfy: closure, associativity, the existence of an identity element, and the existence of inverse elements for each element in the set.*

For instance, a widely studied group is the Euclidean group, which includes translation, rotation, and reflection. Given a group $\mathcal{G}$, the definition of $\mathcal{G} - Equivariance$ is:

**Definition 2.3** *($\mathcal{G}$-Equivariance) A function $f : \boldsymbol{X} \to \boldsymbol{Y}$, where $\boldsymbol{X}, \boldsymbol{Y} \subset \mathbb{R}^2$, is equivariant to group $\mathcal{G}$, if for any transformation $g \in \mathcal{G}$,*

$$f(g \circ x) = g \circ f(x), \quad x \in \boldsymbol{X}. \tag{2}$$

The symmetry of the real world is not always continuous. We focus on discrete groups within finite elements.

## 3 Methodology

### 3.1 Overview

Figure 2 illustrates the overall framework of CrowdEGL. The high-level idea is to learn a latent graph representation that is sensitive to a set of symmetric transformations. To achieve this property, CrowdEGL is additionally trained to predict symmetric transformations of input. As shown in Figure 2, for crowd trajectories of 90°crossings, we consider three more proxy tasks that predict 90°/180°/270°rotations of the original input. Although CorwdEGL can be flexibly adapted to various latent GNNs, to provide sufficient global environmental information, we employed a multi-channel GNN to incorporate original and all augmented input. Finally, to distinguish information learned from the original or augmented tasks, each task has an independent graph decoder to forecast the crowd positions of the next state.

### 3.2 Equivariant Graph Learning

In this section, we introduce how CrowdEGL constructs augmented symmetrical data and learns from them.

**Equivariant data augmentation.** Data augmentation strategies are the core of learning an equivariant model. In our work, these strategies are described by a group, which represents the symmetry of target environments. Particularly, we focus on cyclic groups, whose definition is as follows:

**Definition 3.1** *(Cyclic Group) A cyclic group is a group that can be generated by repeatedly applying a single element.*

Note that the group is closure (the result of two elements within a group is still an element of the group). For example, the cyclic group of 90°rotation consists of four elements - identity and 90°/180°/270°rotations. Although different environments typically have distinct symmetry groups, they can be summarized as following three types:

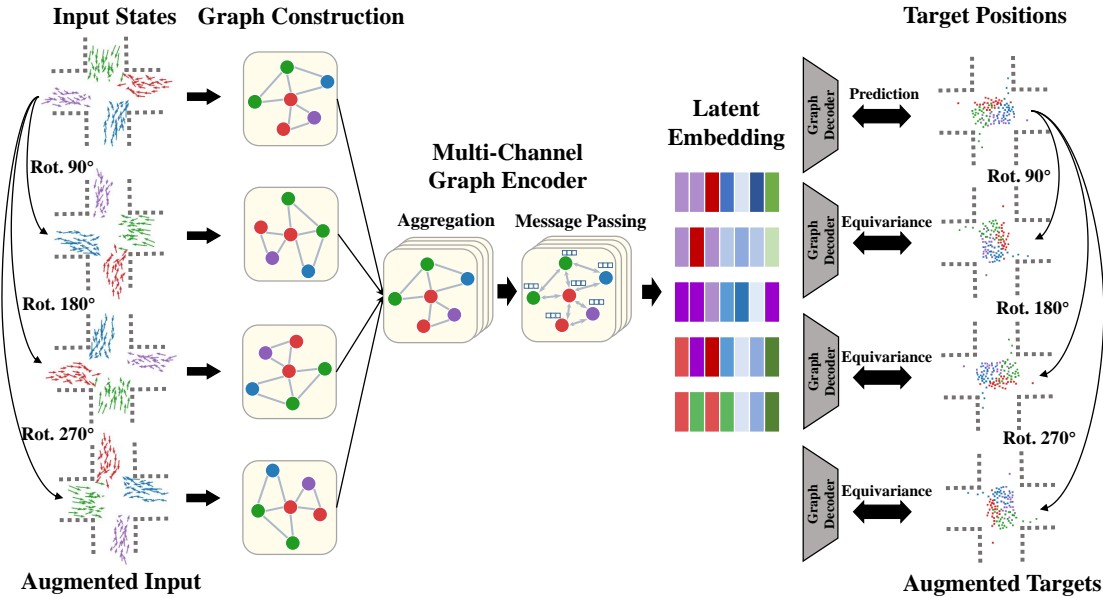

Figure 2: Overview of CrowdEGL framework. Input states and target positions are firstly augmented via group transformations. Then a multi-channel graph encoder is employed to aggregate multi-channel features and perform message passing. Finally, CrowdEGL utilizes multiple independent graph decoders to forecast future positions. The prediction loss and equivariance losses are calculated for the model predictions of original and augmented inputs, respectively.

- **Rotation** is used to augment crowd trajectories in environments with central symmetry, including 90°and 120°crossings.

- **Reflection** is used to construct augmented data for axisymmetric environments. A common example is crowd dynamics in a narrow corridor or T-junction, where we employ a cyclic group of reflections over the x- or y-axis. The intuition is a crowd trajectory from left to right would be similar to that of the same crowd from right to left.

- **Combination of rotation and reflection** is used to generate augmented data for locations like a right-angled corner. In this case, we have to combine reflection and rotation to obtain the reversed trajectory from the exit to the entrance.

**Model training.** The objective function of CrowdEGL consists of a prediction loss and an equivariance loss. Given the group $\mathcal{G}$, the model parameters are optimized by minimizing the following loss $L$:

$$L = L_p + \lambda \frac{1}{|\mathcal{G}|} \sum_{g \in G} L_g, \tag{3}$$

where $L_p$ and $L_g$ denote the prediction loss and equivariance loss of transformation $g$. $|\mathcal{G}|$ is the size of group $\mathcal{G}$. $\lambda$ is a hyper-parameter to control the strength of equivariance loss, a larger $\lambda$ leads to a more strict equivariant model. Since our goal is to forecast the future positions of crowds, models are trained to minimize the discrepancy between the exact and approximated positions. Specifically, Let $f(\cdot; \theta_e)$ be the backbone GNN with parameters $\theta_e$ and $p(\cdot; \theta_p)$ be the decoder for the prediction loss. Then $L_p$ is formulated as:

$$L_p = ||p(f(\boldsymbol{x}^{(t)}, \boldsymbol{v}^{(t)}; \theta_e); \theta_p) - \boldsymbol{x}^{(t+\Delta t)}||_2, \tag{4}$$

where $\boldsymbol{x}^{(t+\Delta t)}$ is the groundtruth. Here we omit the static input $\boldsymbol{u}, \boldsymbol{e}, \boldsymbol{a}$ for simplicity. For equivariance loss, let $\boldsymbol{M}_g \in \mathbb{R}^{2 \times 2}$ be the representation (e.g., rotation or reflection matrix) of the group element $g \in \mathcal{G}$, the equivariance loss $L_g$ is computed by:

$$L_g = ||p(f(\boldsymbol{x}^{(t)}\boldsymbol{M}_g, \boldsymbol{v}^{(t)}\boldsymbol{M}_g; \theta_e); \theta_g) - \boldsymbol{x}^{(t+\Delta t)}\boldsymbol{M}_g||_2. \tag{5}$$

Note that, depending on the backbone GNN, decoder networks for prediction loss can be shared with that of equivariance losses. During experiments, we found that multiple independent decoders achieve better performance.

### 3.3 Backbone GNN

To design a backbone under the CrowdEGL framework, a common solution is to utilize a shared graph encoder and decoder (e.g., GNS (Sanchez-Gonzalez et al., 2020)) for all original and transformed input. However, the crowds may not be exactly symmetric due to the complex human interactions. Simply treating the augmented inputs as original inputs may lead to a suboptimal performance. Instead of using a hard parameter-sharing strategy, we design a flexible multi-channel pipeline to share partial parameters across tasks and learn equivariance from data. Therefore, if the augmented trajectories are exactly equivariant, the CrowdEGL can degrade to the hard parameter-sharing strategies, where the parameters of each channel are the same. Specifically, we first construct a graph and initialize the multi-channel node embeddings based on the defined group. Then multiple steps of message passing are performed to obtain latent graph embeddings. The final predictions are computed via independent graph decoders. We elaborate on the model designs as follows.

**Graph construction.** For personal safety and comfort, pedestrians adjust their trajectories in high-density crowds to avoid collisions with others or obstacles (Yu et al., 2023). Consequently, when they approach each other, pedestrians often instinctively adjust their movement velocities and alter their directions. To model these interactions, we treat all pedestrians and obstacles as nodes and build a radius graph. That is, nodes are connected if the distance between them is smaller than a predefined radius $r$.

**Multi-channel node embedding.** This subsection investigates how to attain desirable initialization for node embeddings that involve 2D symmetry information. To integrate the global environment information, given the rotation matrix representations $\boldsymbol{M}_g$ of group element $g \in \mathcal{G}$, The initial embedding of node $i$ is derived by:

$$\boldsymbol{h}_{i,g} = \phi(\boldsymbol{x}_i \boldsymbol{M}_g, \boldsymbol{v}_i \boldsymbol{M}_g, \boldsymbol{u}_i), \tag{6}$$

where $\boldsymbol{h}_{i,g}$ represents the features under the transformation $g$, which includes both the original (i.e., identity transformation) and augmented inputs. $\phi$ is a linear embedding layer. $\boldsymbol{u}_i$ is the node type embedding, including the pedestrian and obstacle. $\boldsymbol{x}_i, \boldsymbol{v}_i$ are position and velocity features, respectively. The velocities of obstacles are set to zero. Then the initial multi-channel node embeddings $\boldsymbol{h}_i^{(0)}$ are obtained by aggregating all the above features:

$$\boldsymbol{h}_i^{(0)} = \text{Agg}(\{\boldsymbol{h}_{i,g}\}_{g \in \mathcal{G}}). \tag{7}$$

It is worth noting that the aggregation function should not be permutation invariant (Zaheer et al., 2017), such as Mean or Sum. These functions result in a $\mathcal{G}$-invariant GNN and latent embeddings. That is, their outputs remain invariant under input transformations of group $\mathcal{G}$. Consequently, the model can not distinguish the original from augmented inputs and would map all original and augmented inputs to the same position, instead of a $\mathcal{G}$-equivariant model. For CrowdEGL, we use the following aggregation:

$$\boldsymbol{h}_i^{(0)} = \phi_a([\boldsymbol{h}_{i,g}]_{g \in \mathcal{G}}), \tag{8}$$

where $\phi_a$ is an MLP. $[\boldsymbol{h}_{i,g}]_{g \in \mathcal{G}}$ is the concatenated feature that follows a fixed order of group elements.

**Message passing.** In high-density crowds, the interactions among pedestrians, and that between pedestrians and obstacles, are extremely dense and significantly influence their trajectories. Thus the quality of approximating these interactions determines the model performance. We adopt the message-passing framework, which has been proven effective in learning such dynamics. Our Message Passing Layer (MPL) takes

Table 1: Dataset statistics and symmetry property.

| Dataset | #Sample | Mean #Pedestrian | Max #Pedestrian | Target $\Delta t$ | Cyclic group | #Elements |
|---------|---------|------------------|-----------------|-------------------|--------------|-----------|
| **Corner** | 2813 | 13 | 31 | 4 | 90° Rotation and Reflection | 2 |
| **T-Junction** | 5927 | 34 | 70 | 4 | Reflection | 2 |
| **Crossing**-90° | 6581 | 129 | 224 | 20 | 90° Rotation | 4 |
| **Crossing**-120° | 10680 | 132 | 222 | 10 | 120° Rotation | 3 |
| **Corridor**-Uni. | 17231 | 81 | 160 | 6 | Reflection | 2 |
| **Corridor**-Bi. | 10405 | 88 | 143 | 10 | Reflection | 2 |
| **Stadium Gate**-Up | 7461 | 97 | 197 | 10 | Reflection | 2 |
| **Stadium Gate**-Low | 7047 | 103 | 198 | 10 | Reflection | 2 |

the set of node embeddings $\boldsymbol{h}^{(l)}$ ($\boldsymbol{h}^{(0)}$ is initialized via Eq. 8) and edges $\boldsymbol{e}$ as input and outputs a transformation on $\boldsymbol{h}^{(l+1)}$. Concisely, $\boldsymbol{h}^{(l+1)} = \text{MPL}(\boldsymbol{h}^{(l)}, \boldsymbol{e})$. The definition of message embeddings is as follows:

$$\boldsymbol{m}_{ij}^{(l)} = \phi_m(\boldsymbol{h}_i^{(l-1)}, \boldsymbol{h}_j^{(l-1)}, a_{ij}), \tag{9}$$

where $\phi_m$ denotes the MLP. $a_{ij}$ are edge attributes that are the distance $||\boldsymbol{x}_i - \boldsymbol{x}_j||_2$ between node $i$ and $j$. Then the nodes aggregate the message embeddings and update their embeddings as follows:

$$\boldsymbol{h}_i^{(l+1)} = \phi_h(\boldsymbol{h}_i^{(l)}, \sum_{j \in \mathcal{N}_i} \boldsymbol{m}_{ij}^{(l)}) + \boldsymbol{h}_i^{(l)}. \tag{10}$$

Here the node embedding is updated via the skip connection. After $L$ iterations of message passing, we obtain the latent graph embeddings $\boldsymbol{h}_i^{(L)}$.

**Graph decoder**  There are two main designs to predict future positions from $\boldsymbol{h}^{(L)}$. First, most existing models Shi et al. (2023) rely on a simple MLP as the decoder, ignoring crucial interactions in high-density crowds. Hence, $\boldsymbol{h}^{(L)}$ is decoded by several message passing layers:

$$\boldsymbol{x}^{(t+\Delta t)} = \boldsymbol{x}^{(t)} + \text{MPL}(\cdots(\text{MPL}(\boldsymbol{h}^{(L)}, \boldsymbol{e}), \cdots), \boldsymbol{e}). \tag{11}$$

The decoder outputs the change of crowd positions and the skip connection of the decoder's last layer is removed due to the dimension. Second, to increase model capacity and allow models to perform differently for prediction and equivariant tasks, we use independent decoders for each task instead of shared decoders.

## 4 Experiments

### 4.1 Settings

**Datasets**  Methods are evaluated on public pedestrian dynamic data[1] that is built up by the Institute for Advanced Simulation 7:Civil Safety Research of Forschungszentrum Jülich (Cao et al., 2017). The density of these data is extremely high and their scenarios are specially designed for observing crowd dynamics. Dataset statistics are shown in Table 1. The target time interval is determined by the space size and crowd speed. For all datasets, 70%, 10%, and 20% are randomly split over time for training, validation, and testing, respectively. We use the following five environments: (1) **Crossing**: crowds start from all entrances and go through a 90°or 120°crossing; (2) **Corridor**: a crowd goes through a corridor unidirectionally or bidirectionally; (3)**Stadium Gate**: a large number of pedestrians enter or leave the stadium through a gate; (4) **Corner**: a large number of pedestrians move from one side of the corner to the other side; (5)**T-Junction**: two groups of pedestrians entered from both sides of the T-junction and exited from the middle. Note that our work focuses on modeling high-density crowd trajectories where the pedestrian space is severely constrained by others and obstacles. Thus, we do not test our model on traditional human trajectory datasets such as ETH/UCY (Pellegrini et al., 2009; Lerner et al., 2007) and GC (Zhou et al., 2012), where crowds are sparse and more random.

---

[1]https://ped.fz-juelich.de/database/doku.php

Table 2: MSE (in decimeters) of all models. Bold font indicates the best result and Underline is the strongest baseline. We report both mean and standard deviation that are computed over 5 runs. Abs. Improv. denotes absolute improvement.

| Model | Crossing | | Corridor | | Stadium Gate | | Corner | T-Junction |
|---|---|---|---|---|---|---|---|---|
| | 90 Degree | 120 Degree | Unidirectional | Bidirectional | Enter | Leave | | |
| MLP | 16.359± 0.001 | 14.224± 0.001 | 3.283± 0.001 | 5.984± 0.001 | 3.939± 0.001 | 5.622± 0.001 | 1.133± 0.001 | 0.962± 0.001 |
| S-LSTM | 13.690±0.067 | 12.451±0.030 | 2.961±0.003 | 5.116±0.016 | 3.148±0.008 | 4.585±0.039 | 0.943±0.013 | 0.804±0.004 |
| TransF | 12.682± 0.088 | 12.352± 0.030 | 3.061± 0.064 | 5.123± 0.009 | 3.106± 0.026 | 4.515± 0.011 | 0.924± 0.005 | 0.785± 0.005 |
| GNS | 1.609± 0.007 | 2.254± 0.093 | 0.191± 0.009 | 0.443± 0.013 | 0.391± 0.001 | 0.427± 0.011 | $\underline{0.100}$± 0.002 | 0.140± 0.001 |
| EGNN | 3.548± 0.951 | 2.900± 0.284 | 0.248± 0.054 | 0.899± 0.015 | 0.496± 0.030 | 0.612± 0.049 | 1.258± 0.008 | 0.323± 0.001 |
| GMN | 3.668± 0.240 | 3.957± 0.070 | 0.652± 0.129 | 0.789± 0.310 | 0.779± 0.108 | 0.866± 0.018 | 0.545± 0.004 | 0.246± 0.008 |
| CrowdSim | $\underline{1.421}$± 0.043 | $\underline{1.815}$± 0.043 | $\underline{0.167}$± 0.004 | $\underline{0.380}$± 0.019 | $\underline{0.314}$± 0.007 | $\underline{0.413}$± 0.032 | 0.107± 0.007 | $\underline{0.113}$± 0.001 |
| SEGNO | 2.279±0.859 | 5.789±5.311 | 0.694±0.234 | 0.981±1.229 | 0.331±0.210 | 0.415±0.016 | 0.513±0.012 | 0.261±0.059 |
| **CrowdEGL** | **0.530± 0.024** | **0.638± 0.044** | **0.056± 0.001** | **0.121± 0.008** | **0.119± 0.010** | **0.161± 0.009** | **0.073± 0.001** | **0.063± 0.001** |
| **Abs. Improv.** | **0.891** | **1.177** | **0.111** | **0.259** | **0.195** | **0.252** | **0.034** | **0.050** |

**Baselines** We compared CrowdEGL against various baselines including (1) basic MLP; (2) trajectory forecasting models: S-LSTM (Alahi et al., 2016) and TransF (Giuliari et al., 2021); (3) fundamental GNNs: GNS (Sanchez-Gonzalez et al., 2020), and CrowdSim (Shi et al., 2023); (3) equivariant GNNs: EGNN (Satorras et al., 2021), GMN (Huang et al., 2022), and state-of-the-art SEGNO (Liu et al., 2024).

**Evaluation metric** Following previous studies (Shi et al., 2023; Satorras et al., 2021), we use Mean Square Error (MSE) between model predictions and groundtruth to measure the single-step prediction performance. For multi-step predictions, Average Displacement Error (ADE) and Final Displacement Error (FDE) are utilized as metrics. They are the $l_2$ distance of the predicted whole trajectory/endpoint to the ground truth of the whole trajectory/endpoint.

**Parameter settings** We empirically find that the following hyperparameters generally work well, and use them for all datasets: Adam optimizer with learning rate 0.0005, batch size 100, the hidden dimension 64, weight decay $1 \times 10^{-10}$, the message passing layer number 4 and the decoder layer number 2. All models are trained for 5000 epochs with an early stopping strategy of 100. For CrowdEGL, the strength $\lambda$ of equivariance loss is turned from 0.1 to 1 with a step size of 0.1. All models are implemented based on Pytorch and PyG library (Fey & Lenssen, 2019), trained on GeForce RTX 4090 GPU. The cyclic group used in CrowdEGL can be found in Table 1.

### 4.2 Performance Comparison

In this section, we compare our model with baselines.

**Overall performance** Table 2 displays the performance of CrowdEGL and all baselines. From them, we have the following observations:

- GNNs significantly outperform non-GNN models (i.e., MLP, S-LSTM, TransF), indicating the importance of modeling interactions in high-density crowd dynamics. Although EGNN, GMN, and SEGNO consider equivariance property as well, they underperform GNS and CrowdSim. These results show that strict Euclidean group equivariant models are too strong to model the crowd dynamics and even have a negative effect. For example, EGNN underperforms MLP in the Corner dataset.

- CrowdEGL consistently outperforms all baselines under all circumstances in a large gap. In particular, CrowdEGL achieves 1.177 and 0.891 lower MSE than the runner-up model CrowdSim. Such improvement verifies the effectiveness of learning a soft equivariance via data augmentation and a multi-channel GNN backbone.

**Multi-step prediction performance** We additionally evaluate model performance on multi-step predictions as well. Following the previous work(Xu et al., 2023), the decoder is slightly modified to perform

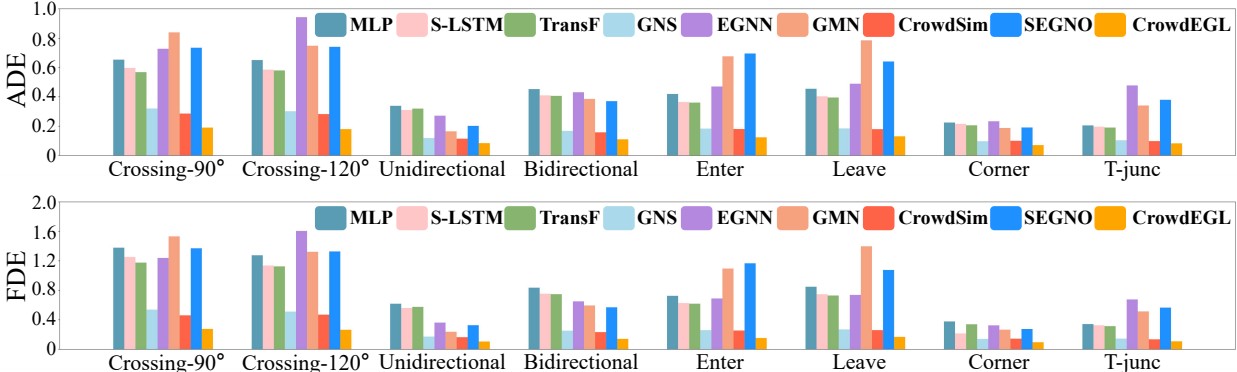

Figure 3: MSE (in decimeters) of multi-step predictions. CrowdEGL achieves the best performance.

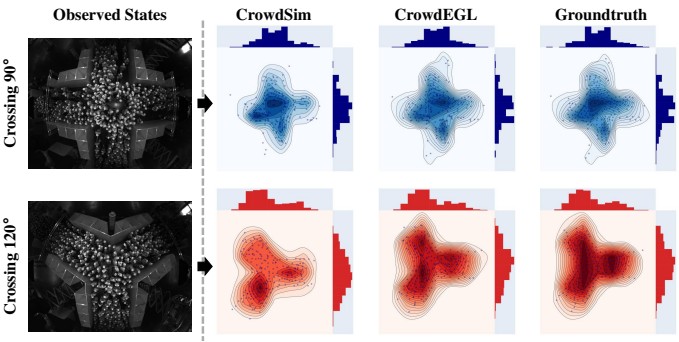

Figure 4: Visualization of CrowdSim and CrowdEGL prediction of high-density crowds. Our method produces a crowd distribution that is closer to the groundtruth.

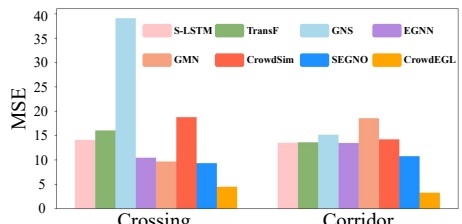

Figure 5: Generalization experiments. **90°Crossing**: models are trained on trajectories from the right and bottom entrances and tested on those from the left and top. **Corridor**: models are trained on left-to-right trajectories and tested in the reversed direction.

multistep predictions. We set the timestep $\Delta t$ to 1 and aim to predict the future 10 steps. The results are displayed in Figure 3. We can observe that CrowdEGL consistently outperforms baselines with the lowest ADE and FDE, demonstrating its effectiveness.

**Visualization**. To gain more insights into the superior performance of CrowdEGL, the density of predicted crowd positions of CrowdSim and CrowdEGL are visualized in Figure 4. We choose samples that have a high density. Particularly, the pedestrian numbers are 199 and 217 in the case of Crossing 90°and 120°. According to the visualization results, we can conclude that CrowdEGL accurately predicts the distribution of all directions, while CrowdSim only captures the patterns in partial directions. For example, the x-axis of CrowdSim output is similar to the ground truth, but it has an obvious difference on the y-axis. Such results again demonstrate the effectiveness of our equivariant training framework.

**Generalization ability**   To further investigate whether the proposed model learns the symmetry of crowd dynamics, we conduct a generalization experiment on Corridor and 90°Crossing environments. For Corridor, models are trained on left- to-right trajectories and tested in the reversed direction. Similarly, in 90°Crossing, models are trained on trajectories from the right and bottom entrances and tested on those from the left and top. The results are shown in Figure 5 where the results of MLP are omitted since they are too high. Under such a situation, the performance of all baselines decreases by a large margin. In contrast, the test error of CrowdEGL is still at a relatively low level.

**Running time**   To compare the efficiency, we report the running time of all methods in the largest two datasets (i.e., Crossing-120° and Corridor-Unidirectional) in Table 4. Results are run on GeForce RTX 4090 GPU. From the table, we can find that CrowdEGL has a comparable efficiency with CrowdSim.

Table 3: Ablation Studies (in decimeters) on model designs. Results are computed over 5 runs. Rel. Improv. denotes relative improvement.

| Model | Crossing | | Corridor | | Stadium Gate | | Corner | T-Junction |
|---|---|---|---|---|---|---|---|---|
| | 90 Degree | 120 Degree | Unidirectional | Bidirectional | Enter | Leave | | |
| Only data augmentation | 2.123±0.252 | 1.989±0.126 | 0.155±0.009 | 0.195±0.018 | 0.331±0.009 | 0.385±0.018 | 0.156±0.001 | 0.163±0.001 |
| Shared decoder | 1.494± 0.088 | 1.341± 0.031 | 0.101± 0.006 | 0.192± 0.011 | 0.204± 0.000 | 0.304± 0.024 | 0.114± 0.003 | 0.098± 0.001 |
| MLP decoder | 0.946± 0.001 | 1.010± 0.105 | 0.089± 0.004 | 0.201± 0.024 | 0.198± 0.000 | 0.207± 0.006 | 0.097± 0.001 | 0.088± 0.001 |
| w/o Multi-channel | 0.713± 0.001 | 0.743± 0.030 | 0.061± 0.003 | 0.160± 0.070 | 0.147± 0.007 | 0.199± 0.011 | 0.080± 0.001 | 0.066± 0.001 |
| $\lambda = 0$ | 0.583± 0.086 | 0.730± 0.026 | 0.063± 0.005 | 0.130± 0.019 | 0.124± 0.004 | 0.186± 0.027 | 0.081± 0.001 | 0.065± 0.001 |
| Default | **0.530± 0.024** | **0.638± 0.044** | **0.056± 0.001** | **0.121± 0.008** | **0.119± 0.010** | **0.161± 0.009** | **0.073± 0.001** | **0.057± 0.001** |
| Rel. Improv. | **9.09%** | **12.60%** | **8.20%** | **6.92%** | **4.03%** | **13.44%** | **9.88%** | **12.31%** |

Table 4: Running time (in seconds) comparison on the largest two datasets. Results are run on GeForce RTX 4090 GPU. Uni. is short for Unidirectional.

| Dataset | MLP | S-LSTM | TransF | GNS | EGNN | GMN | CrowdSim | SEGNO | CrowdEGL |
|---|---|---|---|---|---|---|---|---|---|
| **Crossing-120°** | 1.493 | 2.190 | 3.150 | 4.608 | 5.513 | 4.534 | 3.331 | 3.136 | 3.643 |
| **Corridor-Uni.** | 2.000 | 2.583 | 3.217 | 3.220 | 3.886 | 4.004 | 3.060 | 3.428 | 3.280 |

## 4.3 Ablation Study

**Model design** To evaluate the benefits of our model designs, we compare CrowdEGL with five model variants: (1) **Only data augmentation**: simply treat augmented data as observed trajectories to train the GNN (without multi-channel and independent graph decoder); (2) **Shared decoder**: use the same decoder for both prediction and equivariance tasks; (3) **MLP decoder**: replace graph decoders with MLPs; (4) **w/o Multi-channel**: removes multi-channel inputs and only employ the original input features as initialize node embedding; (5) $\lambda = 0$: remove equivariance loss by setting the hyperparameter $\lambda$ to zero. Table 3 shows the results of all models. We have the following findings:

- The straightforward solution, only employing data augmentation, underperforms all model variants, indicating it is insufficient to model the challenging crowd dynamics. Utilizing a shared decoder in CrowdEGL leads to a significant drop in performance, mainly because the models fail to distinguish the observed or augmented data, reducing the quality of learned latent graph embeddings.

- MLP decoders underperform graph decoders. These observations suggest that modeling interactions are essential in not only the encoding but also decoding processes, especially in the high-density case where interactions are frequent and dense. Removing multi-channel information results in a local model and enhances model errors, demonstrating that explicitly integrating the symmetric features in the encoder can learn a better latent representation.

- Incorporating EGL largely improves the model performance in various environments by encoding equivariance in latent representations. Specifically, EGL achieves 12.60%, 13.44%, and 12.31% relative improvement on Crossing, Stadium Gate, and T-Junction datasets, respectively. Such results demonstrate its effectiveness.

**Performance w.r.t. crowd density** To investigate the model performance on crowd density, we divide the testing sample into 4 equal intervals according to the number of pedestrians. The results are shown in Table 5. From the results, we can observe that the MSE decreases as the density increases, validating our motivation.

**Model performance w.r.t.** $\lambda$ It is crucial to control the degree of equivariance for real-world crowd dynamics modeling. The sensitivity of $\lambda$ on Crossing and Corridor datasets is illustrated in Figure 6. According to the results, we can find that its values largely influence the model performance. A larger value first leads to a lower error, and then enhances the error, indicating strict equivariance does not achieve better performance for real-world crowd dynamics.

Table 5: Mean square errors w.r.t. crowd density. Testing samples are divided into 4 equal intervals according to the number of pedestrians. Uni. and Bi. denote Unidirectional and Bidirectional.

| Density | Crossing-90° | | | | Crossing-120° | | | | Corridor-Uni. | | | | Corridor-Bi. | | | |
|---|---|---|---|---|---|---|---|---|---|---|---|---|---|---|---|---|
| | 42-95 | 96-149 | 150-206 | 207-262 | 126-179 | 180-233 | 234-287 | 288-343 | 49-88 | 89-128 | 129-170 | 171-212 | 41-74 | 75-108 | 109-143 | 144-179 |
| MSE | 2.371 | 0.710 | 0.570 | 0.384 | 6.009 | 1.138 | 0.532 | 0.343 | 0.184 | 0.069 | 0.053 | 0.039 | 0.792 | 0.149 | 0.102 | 0.079 |

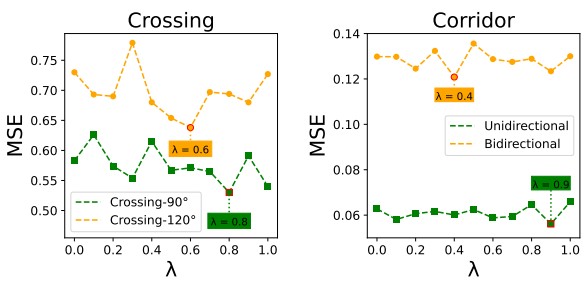

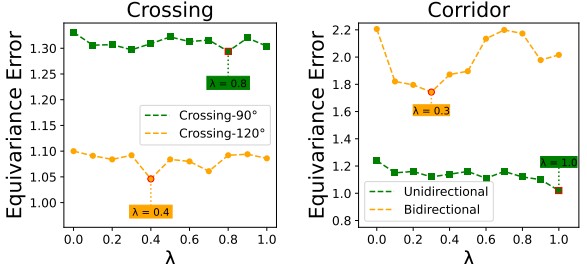

Figure 6: Model performance w.r.t. $\lambda$ on Crossing and Corridor.

Figure 7: Equivariance error w.r.t. $\lambda$ on Crossing and Corridor. We scale the equivariance error of the Corridor-Unidirectional by a factor of 10.

**Multi-channel aggregation functions** The choice of multi-channel aggregation functions is the basis for constructing an equivariant model. We replace the non-linear transformation of multi-channel inputs in Eq.8 by Mean and Sum pooling and show their results on Crossing and Corridor in Figure 8. The key finding is that these permutation invariant functions will significantly reduce model accuracy since they make the model invariant to group transformation. Such invariance contradicts the equivariant nature of crowd dynamics, leading to poor performance.

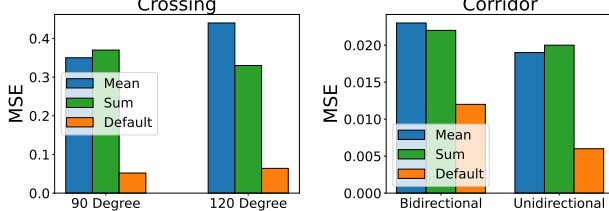

Figure 8: Performance w.r.t. the multi-channel aggregation function.

**Equivariance error w.r.t. $\lambda$** To provide insights into the imperfect equivariance of models, we can computer the equivariance error of group $g \in \mathcal{G}$ as:

$$L_{eq} = ||f(g \circ x) - g \circ f(x)||_2. \tag{12}$$

A smaller error indicates the model is more equivariant. If the model is exactly equivariant, the equivariance error will be zero. We compute the average model equivariance errors of considered groups and display the results in Figure 7. From the figure, we can find that employing equivariant training generally achieves lower equivariance errors. A larger $\lambda$ generally leads to a lower equivariance error in Crossing-90° and Corridor-Unidirectional, while in Crossing-120° and Corridor-Bidirectional, the lowest equivariance errors are achieved at around 0.4.

# 5 Related Work

## 5.1 Modeling crowd dynamics

Traditional methods simulate pedestrian behaviors, especially collision avoidance, based on expert knowledge. Typical methods include rule-based (Reynolds, 1987), force-based (Helbing & Molnar, 1995; Saboia & Goldenstein, 2012), and velocity-based models (Fiorini & Shiller, 1998). Nevertheless, the designed formulas are insufficient to model complex and uncertain real-world dynamics. With the development of deep learning, recent studies (Shi et al., 2023; Yu et al., 2023; Zhang et al., 2022) seek neural solutions to improve

simulation performance. Another similar research line is pedestrian trajectory prediction (Alahi et al., 2016; Gupta et al., 2018), whose goal is to predict pedestrians' future positions based on their historical trajectories. However, they generally study the pedestrian trajectories in urban open spaces where interactions are relatively sparse. Thus, they focus on capturing their sequential dependency via time-series models, such as recurrent models (Alahi et al., 2016; Feng et al., 2018) and Transformer (Yu et al., 2020). Different from the above studies, our target is to predict the future position of high-density crowds based on current states. In this task, interactions are extremely dense and significantly affect crowd trajectories. Thus, we focus on modeling such interactions via GNNs.

### 5.2 Graph Neural Netwoks

**Equivariant graph neural networks**   Recently, to model the dynamics of particle systems (Huang et al., 2022), researchers have proposed equivariant GNNs that meet specific symmetry constraints. They design provably equivariant message passing layers, which could be broadly classified into two types (Han et al., 2022a). First, leverage irreducible representation to denote higher-order geometric features, such as TFN (Thomas et al., 2018; Fuchs et al., 2020) and SEGNN (Brandstetter et al., 2022). Second, construct an invariant scalarization message embedding like the inner product (Satorras et al., 2021; Huang et al., 2022; Zheng et al., 2024). However, they can not be directly extended to pedestrian dynamics. Because of human psychology and physiology, crowd trajectories might not be exactly the same after transformation. Furthermore, due to the boundary conditions, not all rotation/reflection/translation are valid in environments. In this work, we propose to encourage the equivariance of backbone GNNs via self-supervised training, which can flexibly control the degree of equivariance during the learning procedure. In contrast to the above works, we study equivariant GNNs of a finite group.

**Graph network-based simulators**   Due to the simplicity and effectiveness of graph networks, plenty of studies (Sanchez-Gonzalez et al., 2020; Shi et al., 2023; Li et al., 2019; Allen et al., 2022; Pfaff et al., 2021) have adopted them to simulate complex physical dynamics, including fluids, meshes, and rigid objects (Li et al., 2019). They follow an encoder-processor-decoder framework. The core idea is to use a particle representation of objects, then dynamically construct interaction graphs among them and perform multiple steps of information propagation to model their interactive forces. Despite their promising empirical results, these works have not adequately considered the symmetry of real-world crowd dynamics. As a result, they fail to achieve satisfactory generalization ability to forecast crowd trajectories.

## 6   Conclusion and Future Work

In this work, we propose a novel CrowdEGL for modeling the trajectories of high-density crowds in various environments. It learns a soft equivariant model via data augmentation and an additional equivariance loss. We further design an advanced backbone to incorporate multi-channel inputs and employ independent graph decoders to distinguish original and augmented tasks. Extensive experiments conducted on high-density crowd trajectories in various environments demonstrate that CrowdEGL outperforms all competing methods. Ablation studies have further substantiated the effectiveness of our model designs. For future work, For future work, we will enhance CrowdEGL by designing multi-channel embedding without specific input orders and generalize it to multi-environment scenarios instead of learning on a single environment.

### Acknowledgments

This work was supported by NSFC Grant No. 62206067, HKUST-HKUST(GZ) 20 for 20 Cross campus Collaborative Research Scheme C019 and Guangzhou-HKUST(GZ) Joint Funding Scheme 2023A03J0673.

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
