# OpenReview forum: "Equivariant Graph Learning for High-density Crowd Trajectories Modeling"
_TMLR — Accepted by TMLR_

### Review · Reviewer_SVBZ · 2024-07-18

**Summary Of Contributions:**

This paper introduces a novel Equivariant Graph Learning method, called CrowdEGL, for modeling high-density crowd trajectories in urban environments. Unlike traditional methods that rely on rigid formulas, CrowdEGL uses data-driven models and incorporates symmetry transformations to enhance prediction accuracy. The framework employs a multi-channel GNN to learn latent pedestrian patterns and uses multiple independent graph decoders to predict future positions. Extensive experiments demonstrate that CrowdEGL significantly outperforms existing models.

**Audience:**

Yes

**Claims And Evidence:**

Yes

**Requested Changes:**

See weakness above.

**Strengths And Weaknesses:**

Strengths:
1. The paper introduces CrowdEGL, a method incorporating imperfect equivalence for high-density crowd trajectory prediction.
2. CrowdEGL significantly outperforms state-of-the-art models in generalization and performance on diverse datasets.

Weakness:
1. Motivation:
- The trajectory evolution rules for various crowded trajectories are similar. Hence, the original trajectories and transformed trajectories should adhere to the same trajectory evolution rule. Considering this observation, I believe that a model trained with data augmentation can acquire the shared trajectory evolution rule from crowd trajectories that possess inherent symmetry. It is difficult to comprehend what is learned in CrowdEGL within this paper. If the graph encoder in CrowdEGL can grasp the common trajectory evolution rule, then so can the GNN trained with data augmentation.

2. Method:
- The paper only focuses on a limited set of input transformations, such as rotations of 90, 180, and 270 degrees. I believe that inputs rotated at other angles are also significant.

3. Experiments:
- The robustness of (8) needs further verification. For instance, it is unclear how the performance will be affected when treating the 90/180/270 degree rotated input as the original input. There is a potential risk that CorwdEGL may be overfitted to a specific input order in (8).
- How to measure the density of the crowd, hows the performance changes w.r.t density of the crowd?

---

> ### Author Response · Authors · 2024-08-09
>
> Thanks for your constructive comment. We address your concerns in the following.
>
> >Weakness 1: It is difficult to comprehend what is learned in CrowdEGL within this paper. If the graph encoder in CrowdEGL can grasp the common trajectory evolution rule, then so can the GNN trained with data augmentation.
>
> We thank the reviewer for raising the comparison with GNN trained with data augmentation, which treats the equivariant augmented input as the original input. In contrast, CrowdEGL distinguishes these two types of inputs and treats the equivariant inputs as side information. The intuitions are as follows:
>
> * The real-world trajectory evolution rule is complex.  The original trajectories and transformed trajectories may not share exactly the same evolution rules.
> * GNN trained with data augmentation could be viewed as a special case of CrowdEGL, where the model parameters of all tasks are the same. Instead of using such a hard parameter-sharing strategy, CrowdEGL allows the parameters to be different across tasks and learns them from data. Thus, if the trajectories share the same evolution rule, the CrowdED can degrade to GNN trained with data augmentation.
>
>
> >Weakness 2: The paper only focuses on a limited set of input transformations, such as rotations of 90, 180, and 270 degrees. I believe that inputs rotated at other angles are also significant.
>
> Thanks for your suggestion. To evaluate the significance of other angles, we further add experiments w.r.t. sampling step. For example, by setting the sampling step to 4, we have {0$^\circ$, 90$^\circ$, 180$^\circ$, 270$^\circ$}. The results are as follows:
>
>
> Crossing-90$^\circ$:
> | Sampling Step | 4 | 8 | 12 | 16|
> | :----|:----|:----|:----|:----|
> | MSE  |0.530|0.548|0.594|0.676|
>
> Crossing-120$^\circ$:
> | Sampling Step | 3 | 6 | 9 | 12|
> | :----|:----|:----|:----|:----|
> | MSE  |0.638|0.617|0.697|0.724|
>
> We can observe that enhancing sampling angles does not enhance model performance in most cases.
>
> >Weakness 3: How the performance will be affected when treating the 90/180/270 degree rotated input as the original input?
>
> Thanks for the question. In the ablation study (Table 3), we compare CrowdEGL with GNNs that only employ data augmentation. This method treats the 90/180/270 degree rotated input as the original input. We can observe that CrowdEGL achieves better performance, verifying its effectiveness.
>
> >Weakness 3: There is a potential risk that CorwdEGL may be overfitted to a specific input order in (8).
>
> Thanks for raising this issue. Since the input order is specified manually,  we can address this issue by ensuring that the order of training/validation/testing is the same. We agree that employing a backbone that can generalize to any input order may achieve better performance. We will consider it in future work.
>
> >Weakness 3: How to measure the density of the crowd, hows the performance changes w.r.t density of the crowd?
>
> Thank you for raising this important point. Since the area of each dataset's environment is fixed, we use the number of pedestrians to measure the sample density. In the following, we display the model performance w.r.t. the sample density in the testing set. We divide the testing sample into 4 equal intervals according to their density and show their average MSE as follows:
>
> Crossing-90$^\circ$:
>
> | Density | 42-95 | 96-149 | 150-206| 207-262|
> | :----|:----|:----|:----|:----|
> | MSE |2.371|0.710|0.570|0.384|
>
> Crossing-120$^\circ$:
>
> | Density | 126-179 | 180-233 | 234-287 | 288-343|
> | :----|:----|:----|:----|:----|
> | MSE |6.009|1.138|0.532|0.343|
>
>
> Corridor-Unidirectional:
>
> | Density | 49-88 | 89-128 | 129-170 | 171-212|
> | :----|:----|:----|:----|:----|
> | MSE |0.184|0.069|0.053|0.039|
>
>
> Corridor-Bidirectional:
>
> | Density | 41-74 | 75-108 | 109-143 | 144-179|
> | :----|:----|:----|:----|:----|
> | MSE |0.792|0.149|0.102|0.079|
>
> From the results, we can observe that the MSE decreases as the density increases.

---

### Review · Reviewer_vwt6 · 2024-07-19

**Summary Of Contributions:**

This paper presents an interesting perspective that the equivalence of crowds should be explicitly addressed and introduces the algorithm named CrowdEGL to enhance the generalization ability of GNNs. In my opinion, the core contributions of this paper include:
1. Emphasizing the importance of explicitly handling equivalence in the field of crowd trajectory prediction using GNNs.
2. Utilizing data augmentation combined with a multi-channel network structure for training.
3. Conducting extensive experiments to validate their methods across different datasets, especially in high-density crowd scenarios.

**Audience:**

Yes

**Broader Impact Concerns:**

Not applied.

**Claims And Evidence:**

Yes

**Requested Changes:**

1. I think Fig. 2 should provide more insights about the multi-channel network structure.
2. It is not necessary to emphasize that designing a backbone GNN is "non-trivial" at the beginning of section 3.3.
3. Please add more experiments and insights on the "imperfectness" of equivariance. For example, how can it be measured in datasets and how to tune it during training? Since the authors agree that it is an important factor influencing results, as illustrated in Figure 6, more detailed analysis would be beneficial.

**Strengths And Weaknesses:**

From my perspective, the strong aspects of this paper include:
1. Novelty in the question set. As mentioned in the paper, considering equivariance in the combined field of crowd trajectory prediction and GNNs is relatively new.
2. Better performance compared to previous methods.

Weaknesses include:
1. Limited novelty in methods. The data augmentation techniques mentioned, such as rotation and reflection, are straightforward and common in machine learning. While the multi-channel network structure is more novel, it requires more detailed explanation and clarification on the types of problems this network structure may address. The intuition behind this structure is not clearly explained in the paper.
2. The evaluation metrics used in the paper, focusing on the MSE of pedestrians over a longer horizon, are similar to the missing rate used in other trajectory prediction papers. However, solely using the missing rate is insufficient for evaluating a prediction model. Given the principal aim of this work is to model interactions among pedestrians and between them and the environment, as stated in the conclusion, outputting a whole trajectory with smaller time steps and evaluating the results using various metrics such as minADE, Missing Rate, minFDE, etc., would provide a better assessment of how well interactions are modeled.

---

> ### Author Response · Authors · 2024-08-09
> **Author response - Part 1**
>
> Thanks for your constructive comment. We address your concerns in the following.
>
> >Weakness 1: Limited novelty in methods.
>
> We are thankful for your positive comments regarding the multi-channel networks. We would like to emphasize our method contributions as follows:
>
> * Our goal of leveraging equivariance in learning high-density crowd dynamics is novel and has never been explored before as far as we know. We show that training with equivariant augmented inputs enhances model performance.
> * We propose a novel pipeline that is mainly composed of three stages: data augmentation with the cyclic group, a multi-channel graph encoder to map original inputs and augmented inputs to a latent embedding, and independent graph decoders.
>
> >Weakness 1: While the multi-channel network structure is more novel, it requires more detailed explanation and clarification on the types of problems this network structure may address. The intuition behind this structure is not clearly explained in the paper.
>
> Thanks for the comment. We address the concerns as follows:
>
> * **Problems**: Due to the complex human interactions, the crowds may not exactly symmetry. Thus, our goal is to encourage the model to be "soft" equivariant via leveraging the symmetric transformation.
> * **Intuition**: Instead of using a hard parameter-sharing strategy, multi-channel networks aim to share partial parameters across tasks and learn equivariance from data. Such a framework is more flexible. If the augmented trajectories are exactly equivariant, the CrowdEGL can degrade to the hard parameter-sharing strategies, where the parameters of each channel are the same.
>
> We have updated the details in Section 3.3.
>
> >Weakness 2: Evaluation of whole predicted trajectories with smaller timesteps.
>
>
> Thank you for providing evaluation metrics. Since we consider the deterministic prediction, we further compare the ADE/FDE of all models. Following existing works[1], we slightly modify the decoder to perform multiple step predictions. We set the timestep $\Delta t$ to 1 and aim to predict the future 10 steps. The results are as follows:
>
> ADE of all models:
>
> | Model |  Crossing-90$^\circ$  |Crossing-120$^\circ$| Unidirectional  | Bidirectional| Enter  |Leave|Corner  |T-junc|
> | :----|:----|:----|:----|:----|:----|:----|:----|:----|
> | MLP     |0.65354|0.65077|0.33847|0.45258|0.41920|0.45492|0.22442|0.20491|
> |S-LSTM   |0.59712|0.58500|0.31019|0.40974|0.36422|0.40356|0.21552|0.19606|
> |TransF   |0.56833|0.57985|0.31957|0.40634|0.36044|0.39515|0.20554|0.19020|
> | GNS     |0.32003|0.30161|0.11867|0.16752|0.18315|0.18429|0.09702|0.10358|
> | EGNN    |0.72775|0.94328|0.27118|0.43119|0.47025|0.48929|0.23314|0.47757|
> |GMN      |0.83950|0.74837|0.16458|0.38542|0.67627|0.78519|0.18722|0.34059|
> |CrowdSim |0.28563|0.28226|0.11417|0.15751|0.18015|0.17931|0.09992|0.09777|
> |SEGNO    |0.73476|0.74102|0.20172|0.37033|0.69544|0.64080|0.19073|0.37950|
> |CrowdEGL |**0.18988**|**0.17955**|**0.08383**|**0.10990**|**0.12365**|**0.13037**|**0.07050**|**0.08241**|
>
>
> FDE of all models:
>
> | Model |  Crossing-90$^\circ$  |Crossing-120$^\circ$| Unidirectional  | Bidirectional| Enter  |Leave|Corner  |T-junc|
> | :----|:----|:----|:----|:----|:----|:----|:----|:----|
> | MLP     |1.37910|1.27644|0.61863|0.83626|0.72569|0.84897|0.37811|0.34272|
> |S-LSTM   |1.25271|1.13673|0.56056|0.75411|0.62765|0.74750|0.21552|0.32610|
> |TransF   |1.17665|1.12550|0.57583|0.74848|0.61891|0.73011|0.33978|0.31428|
> | GNS     |0.53815|0.51173|0.17258|0.25297|0.26003|0.26939|0.14030|0.14473|
> | EGNN    |1.23979|1.60635|0.36219|0.65059|0.69000|0.73834|0.32489|0.67660|
> |GMN      |1.53274|1.32323|0.23788|0.59485|1.09593|1.39819|0.26637|0.51428|
> |CrowdSim |0.46075|0.47027|0.16428|0.23304|0.25484|0.25997|0.14407|0.13549|
> |SEGNO    |1.37099|1.32815|0.32695|0.57002|1.16732|1.07651|0.27556|0.56562|
> |CrowdEGL |**0.27646**|**0.26449**|**0.10533**|**0.14143**|**0.15325**|**0.16812**|**0.09592**|**0.10814**|
>
> We can observe that CrowdEGL consistently outperforms baselines, demonstrating its effectiveness. We sincerely thank the reviewer again for motivating us to evaluate multi-step predictions.
>
>
> [1] EqMotion: Equivariant Multi-agent Motion Prediction with Invariant Interaction Reasoning.
>
>
> >Requested Changes 1: Fig. 2 should provide more insights about the multi-channel network structure.
>
> Thanks for the suggestion. We have updated Fig.2 in the revision, adding the graph construction, multi-channel aggregation, and message passing step.
>
> >Requested Changes 2: It is not necessary to emphasize that designing a backbone GNN is "non-trivial".
>
>  Thanks for the comment. We have improved the writing.

---

> ### Author Response · Authors · 2024-08-09
> **Author response - Part 2**
>
> >Requested Changes 3: More experiments and insights on the "imperfectness" of equivariance.
>
> Thanks for your suggestions. In general, we can compute equivariance error to measure the equivariance of datasets and models, which is defined as follows:
> $$L_{eq}=||f(g\circ x) - g \circ f(x)||_2$$
> where $g\in \mathcal{G}$ is an element of group $\mathcal{G}$.
>
> However, since our datasets are collected from the real world instead of simulation data with known generation rules, we can not compute the equivariance errors of the datasets. We compute the average model equivariance errors of considered groups. The results of varying hyperparameter $\lambda$ are as follows:
>
> | $\lambda$ | 0 | 0.1 | 0.2 |0.3|0.4|0.5|0.6 | 0.7 | 0.8 |0.9|1|
> | :----|:----|:----|:----|:----|:----|:----|:----|:----|:----|:----|:----|
> |  Crossing-90$^\circ$ | 1.331 | 1.306|1.307|1.297|1.309|1.322 |1.313 |1.316|**1.294**|1.320|1.304|
> |  Crossing-120$^\circ$ |1.100 |1.091 | 1.084|1.092|**1.046** |1.084 |1.080|1.061 |1.092 |1.094 |1.086|
> |  Unidirectional|0.124 | 0.115| 0.116|0.112 |0.114 | 0.116| 0.111|0.116 |0.112 |0.110 |**0.102**|
> |  Bidirectional|2.206|1.821 | 1.796|**1.743**| 1.872|1.896 | 2.135|2.199 |2.173 |1.978 |2.016|
>
> We can find that employing equivariant training generally achieves lower equivariance errors and we can tune $\lambda$ to adjust the model equivariance error.

---

> > ### Comment · Reviewer_vwt6 · 2024-08-12
> >
> > Thank you for your responses. I think my concerns are addressed.

---

> > > ### Author Response · Authors · 2024-08-13
> > >
> > > Thanks for your kind reply.

---

### Review · Reviewer_YJ7y · 2024-07-30

**Summary Of Contributions:**

The work aims to predict the dynamics of high-density crowds for applications in urban planning and architectural design. In particular, this work proposes to use an equivariant approach to exploit (imperfect) symmetries that the crowd dynamics may obey such as e.g. left-to-right vs right-to-left symmetry and rotational symmetry. This is done via data augmentation combined with a weighted symmetry-aware loss. The work shows that this outperforms non-GNN, non-symmetry-aware and data-augmentation-only baselines. It also studies the various contribution through ablation studies.

**Audience:**

Yes

**Claims And Evidence:**

Yes

**Requested Changes:**

- "For simplicity, we denote (x(t), v(t)) and (u, e = {eij}, a = {aij}) as dynamic and static state information of the entire graph correspondingly. " is it not clear to me why position would be dynamic, but the distance between nodes would be static, if the position changes, so would the distance and the graph connectivity?

- "The symmetry of real world is generally not continuous." this statement is clearly not true, large parts of physics are based on continuous symmetry such as the entire universe around us obeying E(3) symmetry.

- eqn 5 should be L_g instead of L_p

- " First, most existing models Shi et al. (2023) rely on a simple MLP as the decoder, ignoring crucial interactions in high-density crowds. " --> it's not really clear why going from a simple MLP to the iterated MLP shown in the eqn would make such a big difference and why a simple MLP on the already learned, highly nonlinear latents would "ignore crucial interactions"?

- how are the data split, if they are temporal are they split s.t. earlier segments are train, later are val/test or are they split randomly over time?

**Strengths And Weaknesses:**

The work shows strong improvements with a fairly simple idea. The ablation studies are well-designed and answer the questions a reader may have about the different subparts of the system. The paper is a nice progress.

One small weakness is that it would've been nice to see a comparison of the extra compute required at training time for the data augmentation / equivariant setup vs the vanilla GNN baseline.

---

> ### Author Response · Authors · 2024-08-09
>
> Thanks for your constructive comment. We address your concerns in the following.
>
> >Weakness: Comparison of model training time.
>
> Thanks for your suggestion! In the following table, we report the training time (in seconds) of all methods in the largest two datasets. Results are run on GeForce RTX 4090 GPU and averaged over five runs. The results show that CrowdEGL has a comparable efficiency with CrowdSim.
>
> | Model | Crossing-120$^\circ$ | Corridor-Unidirectional |
> | :----|:----|:----|
> | MLP |  $1.493$ | $2.000$ |
> |S-LSTM| $2.190$ | $2.583$ |
> |TransF| $3.150$ | $3.217$ |
> | GNS | $4.608$ | $3.220$ |
> | EGNN | $5.513$ | $3.886$ |
> |GMN |$4.534$ | $4.004$ |
> | CrowdSim | $3.331$ | $3.060$ |
> | SEGNO | $3.136$ | $3.428$ |
> |CrowdEGL | $3.643$ | $3.280$ |
>
>
> >Requested Changes 1: why position would be dynamic, but the distance between nodes would be static, if the position changes, so would the distance and the graph connectivity?
>
> Apologize for the confusion. Our goal is to forecast the subsequent position of pedestrians given the initial states (Definition 2.1). Since the initial state is a single snapshot of pedestrians, we treat it as a static input. The distance between nodes and graph connectivity is different for distinct initial states. We acknowledge that the current statement may potentially lead to misunderstandings. Thus, we have removed "static" in the definition.
>
> >Requested Changes 2: large parts of physics are based on continuous symmetry.
>
> Thanks for raising this issue. We have changed the statement to "The symmetry of the real world is not always continuous."
>
>
> >Requested Changes 3: eqn 5 should be L_g instead of L_p.
>
> Thanks for the comment. We have corrected the Eq.5.
>
>
> >Requested Changes 4: why going from a simple MLP to the iterated MLP shown in the eqn would make such a big difference and why a simple MLP on the already learned, highly nonlinear latents would "ignore crucial interactions"?
>
>
> Thanks for the questions. We would like to clarify that:
>
> * Latent embeddings are not already learned. CrowdEGL is an End-to-End learning method, which updates latent embeddings during training. Thus, a well-designed decoder is vital for learning crowd interactions.
> * Pedestrian interactions in high-density crowds are highly frequent. Compared with simple MLP, employing message passing layer (MPL) leverages additional relational information which is beneficial to learning pedestrian interactions.
>
> >Requested Changes 5: how are the data split, if they are temporal are they split s.t. earlier segments are train, later are val/test or are they split randomly over time?
>
> Thanks for the question. We split datasets randomly over time.

---

> > ### Comment · Reviewer_YJ7y · 2024-08-13
> >
> > The authors have address my remaining, small concerns.

---

> > > ### Author Response · Authors · 2024-08-14
> > >
> > > Thanks for your kind reply.

---

### Author Response · Authors · 2024-08-09
**General response**

Dear Editor and Reviewers,

Thank you for your time in providing valuable feedback on our manuscript. A comprehensive point-by-point response to the reviewers' comments is presented below. We have updated the main manuscript with the revised texts highlighted in blue. The major changes are listed below:

* **Additional empirical evaluation**: We have significantly enhanced the empirical validation of the proposed methodology, including: (1) running time comparison; (2) evaluation of multi-step trajectory prediction; (3) model performance w.r.t. crowd density; (4) analyzing the imperfect equivariance via equivariance errors.
* **Presentation**: We have achieved a series of presentation enhancements, including: (1) providing more insights into model structures in Fig.2; (2) more detailed explanation and insights of method intuition; (3) correcting various typos and incorporating content suggestions provided by the reviewers.

We hope these changes address the concerns raised by the reviewers.

Best,

Authors

---

### Author Response · Authors · 2024-09-29
**Thank you**

Dear Editor and Reviewers,

We sincerely thank you for your time in chairing and reviewing our manuscript. We have updated the camera-ready version and provided a code link.

Best,

Authors

---

### Decision · Action_Editor_8kvr · 2024-08-26

**Recommendation:** Accept as is

**Comment:**

The paper introduces a method for learning equivariant graphs using GNNs. It is applied for modeling crowd trajectories, and its superior performance is shown across 8 datasets and 5 different environments.

Reviewers vwt6 and YJ7y recommend acceptance. Reviewer SVBZ found that a more realistic evaluation is required (see comments above). Despite that, the paper's claims seem robust and supported by the current experiments.

**Audience:**

The paper deals is about modeling graphs with equivariant properties, and it introduces an algorithm to enhance the generalization ability of GNNs in such cases. It is of interest for researchers and practitioners working with GNNs or graph data.

**Claims And Evidence:**

Overall, the paper's claims are sufficiently supported. Reviewer vwt6 found the experiments are robust and the provided perspective is interesting.

Reviewer SVBZ found that a more realistic evaluation is required, as the datasets are highly controlled, making it unclear why data augmentation works differently across scenarios. The reviewer argues that the current evaluation falls short in that it doesn't show the generalization properties to real cases.